# Rheological and Aging Characteristics of Polymer-Modified Asphalt with the Addition of Sulfur †

**Ana Luiza Rodrigues** *[ID]**, Caio Falcão** and **R. Christopher Williams**

Civil, Construction, and Environmental Engineering Department, Iowa State University, Ames, IA 50011, USA; falcao@iastate.edu (C.F.); rwilliam@iastate.edu (R.C.W.)
* Correspondence: analuiza@iastate.edu
† Presented at the Second International Conference on Maintenance and Rehabilitation of Infrastructure Facilities (MAIREINFRA2), Honolulu, HI, USA, 16–19 August 2023.

**Abstract:** Crosslinking agents, notably sulfur, are used in asphalt binder modification to facilitate chemical bonding between polymer chains and the asphalt binder. Despite some prior research indicating the benefits of sulfur crosslinking in enhancing polymer-modified asphalt's (PMA) stability, there is a lack of comprehensive understanding regarding its impact on rheological properties and its anti-aging potential. This study addresses these gaps by thoroughly investigating the effects of varying the sulfur content (ranging from 0.03% to 0.5% by total weight of binder) on PMA's rheological properties. The research assesses the effectiveness of sulfur in enhancing PMA's resistance to aging using various methods, including the Glover-Rowe parameter, FTIR analysis, and the examination of the dynamic modulus and phase angle master curves. The results indicated that the addition of sulfur, particularly up to 0.3%, bumps the high-temperature performance grade by one level, and significantly improves elasticity, allowing the PMA to support heavier traffic without experiencing rutting, all while maintaining resistance to low-temperature cracking. Furthermore, PMA with sulfur demonstrated an increase in resistance to aging, reducing the aging potential by approximately 15% with the best sulfur formulation. This enhanced durability can reduce the frequency of maintenance activities, leading to cost savings, reduced roadwork emissions, and prolonged pavement life.

**Keywords:** asphalt binder; sulfur; polymer-modified asphalt; rheology; aging; performance





## 1. Introduction

The increase in traffic volume and loading of the pavements has led to an increase in the occurrence of distresses and, consequently, the higher maintenance and rehabilitation costs of roads to preserve their pavement quality and guarantee that the pavement meets its design life. In light of these challenges, it has become important to make asphalt mixtures more cost-effective over their service life and adequate for the current traffic demands. One effective approach to enhancing pavement performance is through the modification of the asphalt binder with polymers [1,2].

Polymer-modified asphalt (PMA) is typically produced by incorporating from 3 to 7% of polymer by the total weight of the binder. This blending process is carried out through simple mechanical dispersion under high shear conditions at elevated temperatures [3]. Research has shown that adding polymer to the asphalt binder creates a continuous polymeric network, enhancing its rheological and viscoelastic properties [4–7]. The copolymer elastomer styrene–butadiene–styrene (SBS) block is the most common polymer asphalt modifier. The styrene interacts with nonpolar components associated with maltenes and improves the material's stiffness at low temperatures, making it more resistant to cracking and breaking in those conditions, while the butadiene component contributes to its elasticity, increasing its rutting resistance at high temperatures [8,9].

However, the polymer–asphalt system is unstable at high temperatures, typical of those in which asphalt binders are stored. Crosslinking agents are used to reduce the

phase separation between the polymer and asphalt binder to improve compatibilization. These agents facilitate the connection between the polymer chains to form a polymer network. A more extensive network reduces the risk of separation, resulting in a more homogenous blend and improving its performance [4,8,10–13]. Elemental sulfur, being one of the most widely used crosslinking agents, is inexpensive, abundant, and widely available. It promotes the vulcanization of SB by chemically crosslinking the elastomer through the unsaturated bond of butadiene and chemically connecting polymer and asphalt molecules via sulfide and/or polysulfide bonds [14,15]. Since the sulfur vulcanization of SB is an anti-plasticization procedure, it forms higher-molecular weight networks, which can lead to gel formation. Because of this, it is crucial to fine-tune the sulfur dosage carefully to prevent the occurrence of gelation [16].

Sulfur is largely produced as a byproduct of the petrochemical and oil and gas refining industries [17]. These industries face a challenge known as the sulfur challenge due to the substantial production of sulfur, resulting in large surpluses of this element with no clear application. Repurposing this excess sulfur as a modifier in asphalt to enhance its properties presents a promising solution, making the industry more environmentally friendly and moving it a step closer to a circular economy [18,19]. Furthermore, incorporating sulfur into asphalt binders offers several advantages: it can serve as a partial substitute for asphalt and functions as a mineral filler in the mastic, enhancing the pavement's stiffness [10,20]. Sulfur-modified PMA demonstrates increased durability and resistance to aging, leading to longer-lasting pavements that demand less maintenance and rehabilitation, thus generating significant long-term cost savings [17]. Reduced maintenance activities and improved reliability result in less resurfacing and the decreased use of heavy machinery, ultimately mitigating air pollution and greenhouse gas emissions. Therefore, integrating sulfur into asphalt, especially as a crosslinking agent in PMA, delivers benefits across economic, environmental, and waste management areas.

Despite the advantages of enhancing PMA stability and performance, it is essential to remain mindful of the potential hazards associated with the use of large amounts of elemental sulfur. The presence of hydrogen sulfide ($H_2S$) during the blending of sulfur with asphalt binders poses a risk to human health, as $H_2S$ is highly toxic and can cause respiratory and other health-related issues. Additionally, the emission of sulfur oxides (SOx), such as sulfur dioxide, during this process can contribute to air pollution and have detrimental effects on the environment and public health. To further understand these hazards in the production process of asphalt, the Federal Highway Administration [17] studied the safety and environmental aspects of asphalt binders containing up to 50% sulfur by total weight of the binder, finding that the levels of these hazardous gases were considerably below the maximum allowable concentration during the blending process. Additionally, Zeng and Zhao [21] investigated the escape of sulfur-containing gases in the production of a polymer-modified asphalt with 0.15%wt. of sulfur and concluded that the amount measured is very low and substantially lower than the maximum hazard level.

The use of sulfur as a crosslinking agent in asphalt dates back to 1958, when Welborn and Babashak [22] recognized that sulfur significantly improved the storage stability of rubber or latex-modified asphalt. In 1976, Maldonado [23] patented a process for preparing storage-stable SBS-modified asphalt by adding sulfur, but practical applications were limited initially due to high viscosity. A breakthrough occurred in 1990 with the successful preparation and application of storage-stable PMA blended with sulfur in real paving projects, leading to extensive research on optimizing the processes, enhancing performance, and reducing costs. The idea of sulfur crosslinking polymer chains and forming chemical bonds between the polymer and asphalt through sulfide and/or polysulfide linkages was central to these developments [24–26]. While this concept was inspired by the well-known vulcanization process of unsaturated rubbers, its applicability to PMAs required extensive study and validation. Evidence supporting the crosslinking reaction of polymer and asphalt came from observations of changes in torque during blending [27], curing curves in rheometers [28], and rheological behavior [29].

The aging of asphalt binders is an inevitable process that begins in the mixing phase, continues during placement in the field, and extends throughout the service life of pavements. The aging occurs mainly due to the loss of volatile oily components from the asphalt binder and the chemical transformation of asphalt molecules as they interact with oxygen [18,30]. This aging is a complex phenomenon influenced by various factors, and its effects can significantly impact the overall quality and durability of the pavement [31].

Two major contributors to asphalt aging are thermo-oxidation, which results from the exposure of asphalt to high temperatures and oxygen, and ultraviolet (UV) radiation from the sun, which accelerates aging through a process called photooxidation. Oxidative aging introduces oxygen-containing functional groups into the asphalt, such as ketones and sulfoxides, along with trace amounts of dicarboxylic anhydrides and carboxylic acids [32,33]. This phenomenon enhances intermolecular interactions within the asphalt, intensifying its hardening and adhesion characteristics. Additionally, the accelerated generation of carbonyl and sulfoxide functional groups due to oxidation leads to further stiffening and embrittlement of the asphalt [34–36]. Consequently, this makes the asphalt less responsive to external forces, contributing to the cracking of the oxidized asphalt and reducing its ability to self-heal. In the context of PMAs, these aging mechanisms can cause the degradation of the polymer and polymer network by displaying chain scission, resulting in a reduction in elasticity and, consequently, decreasing the pavement tolerance to damage and resistance to cracking [1,3,37,38]. While some valuable research has addressed aspects of PMA and sulfur vulcanization to enhance its storage stability, there remain notable gaps in the comprehensive characterization of improvements in the rheological properties and anti-aging potential when sulfur is incorporated into asphalt–polymer blends. Despite individual studies, such as Cuciniello [39] and Tang [10], that have indicated that crosslinked polymer-modified binders exhibited a lower oxidative susceptibility than non-cross-linked binders, a holistic understanding of these critical aspects is currently lacking.

The necessity for addressing these research gaps becomes evident when we consider the pivotal role that an asphalt binder's resistance to aging plays in extending the service life of a pavement by reducing the frequency of maintenance and rehabilitation. Hence, this study is designed to bridge these gaps and provide a thorough examination of the effects of elemental sulfur addition and its concentration on PMA's rheological properties. Furthermore, it evaluates the efficacy of sulfur in delaying PMA aging through various means, including the Glover-Rowe parameter, Fourier-transform infrared spectroscopy (FTIR), and the analysis of the dynamic modulus and phase angle master curves.

## 2. Materials and Methods

### 2.1. Materials

The neat asphalt binder used in this study was classified as PG 64-22S, according to Superpave performance grading (PG). Its physical properties, such as softening point and penetration (25 °C), and rotational viscosity (135 °C), are 50.1 °C, 39.3 $10^{-1}$ mm, and 0.4725 Pa-s, respectively. For the polymer modification of the asphalt, a thermoplastic elastomer consisting of a linear diblock copolymer composed of blocks of styrene and butadiene with a polystyrene content of 33% was used. The sulfur used as a crosslinking agent was a minus 100-mesh powder with 99.5% purity.

This study compared the properties of a control sample, a binder modified with sulfur, and five PMAs with different sulfur contents. The preparation of the samples is illustrated in Figure 1. The first step consisted of gradually incorporating 3% SB by total weight into the pre-heated neat asphalt at 180 °C using a high-shear Silverson L5M-A mixer at 4000 rpm. Following the polymer addition, the mixture was mechanically agitated using an IKA RW 20 mixer for 60 min at 1500 rpm to make the blend essentially homogeneous. Subsequently, the PMA was prepared by mixing it with a calculated sulfur content, including ratios of 0.075%, 0.03%, 0.1%, 0.3%, and 0.5% by weight of binder [9,16,24,26,40,41]. These specific ratios were chosen based on pre-screening tests, which indicated their suitability for the study, and the selection aligns with findings from relevant research. This secondary mixing

process was conducted using a mechanical agitator, which operated at a lower shear stress, at a temperature of 160 °C, to simulate a storage tank temperature and allow an effective curing rate, and at a speed of 800 rpm. At 120 °C, the reaction between asphalt and sulfur and SB and sulfur occurs, and, at higher than 140 °C, sulfur is finely dispersed in liquid asphalt as uniformly small particles [42]. The mixing duration for this step was 4 h for effective vulcanization and development [43], but not exceeding 4 h as recommended by the FHWA [17]. For the control sample and SB-modified sample without sulfur, the same process was followed, with the only difference being that sulfur was not used.

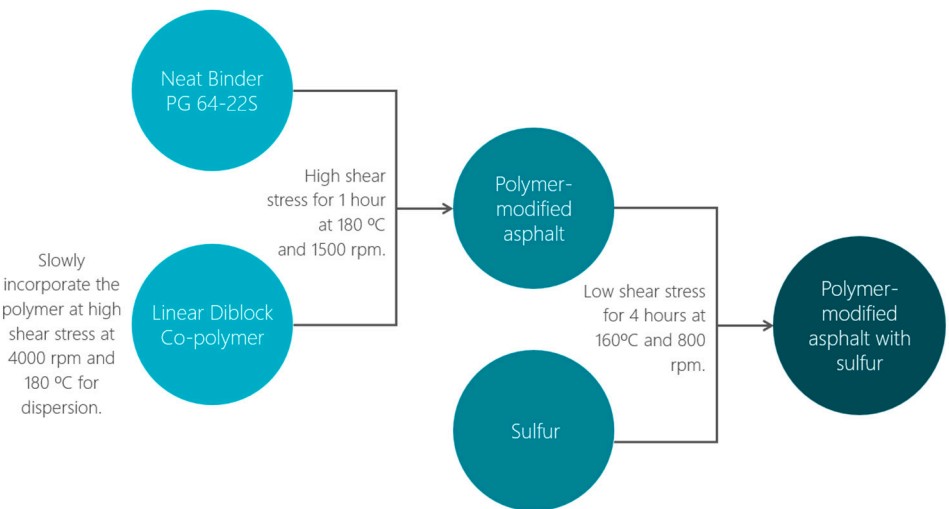

**Figure 1.** Schematic of the preparation of PMA with sulfur.

## 2.2. Methods

Asphalt binders were evaluated based on their Superpave performance grade and rheological properties. The binder grading consists of a set of characterization tests that determine the temperature range in which the binder is suitable to be used in pavement construction. The tests were performed in triplicates and were conducted following the AASHTO M 320 [44] specification, using the dynamic shear rheometer (DSR) Anton Paar Modular Compact Rheometer 302e with parallel plate geometry (25 mm diameter and 1 mm gap, 8 mm diameter and 2 mm gap), and the Cannon TE-BBR SD Thermoelectric Bending Beam Rheometer (BBR), with samples that were aged in the laboratory using the Rolling Thin Film Oven (RTFO), which simulates short-term aging during mixing production, and, subsequently, the Pressure Ageing Vessel (PAV), a process described in AASHTO R 28 [45] that simulates 7 to 10 years of aging due to environmental exposure and traffic load. The Multiple Stress Creep Recovery (MSCR) measures the non-recoverable creep compliance of the binder (Jnr), which can be an indicator of high-temperature rutting sensitivity for unmodified and modified binders, and percent recovery, that can determine and measure the polymer's functionality within the binder. The test is performed on the DSR at the PG temperature, applying a constant stress creep followed by a zero-stress recovery, as indicated in the AASHTO TP 70 [46]. To validate the polymer modification of an asphalt binder and have a high potential to resist permanent deformation at high temperatures, the asphalt binders should have a sufficient delayed elastic response, defined as achieving a minimum %recovery for the measured Jnr.

The rheological master curve represents the asphalt binder characteristic in a viscoelastic region. It is a model based on a time-temperature principle that predicts the material's performance over a range of temperatures and loading times or frequencies. In a master curve analysis of asphalt behavior, the frequency range becomes particularly relevant as it aligns with different traffic conditions and pavement distresses. Lower frequencies (≤0.1 Hz) correspond to slow-moving traffic that experiences prolonged exposure to heavy loads, a scenario characteristic of rutting. Asphalt exhibits pronounced viscous characteris-

tics in this range, expressed by a higher phase angle and reduced stiffness. To resist rutting under such conditions, it is advisable to use a stiffer binder with greater elasticity to resist the deformation of the repeated load and return it to its original state without significant energy dissipation [1,10,47,48]. Conversely, a less rigid and more elastic binder is preferred at higher frequencies (>10 Hz), which relate to shorter loading times with dynamic and impact loads. This flexibility permits deformation without accumulating excessive stresses and the binder can return to its original state without developing cracks [3,48].

In this work, dynamic shear modulus ($|G^*|$) and phase angle ($\delta$) master curves were used to describe the sulfur's influence on the modified asphalt binder's behavior. The binders' $|G^*|$ and $\delta$ values were determined by performing a frequency sweep at various temperatures through the DSR, using the 8 mm parallel plate geometry and 2 mm gap. For each temperature, 16 measurements were obtained by executing a frequency sweep spanning from 100 to 0.1 rad/s [47]. The curves were constructed using the Christensen–Anderson model, and the shift factor was calculated based on the Williams–Landel–Ferry (WLF) time–temperature superposition principle for a reference temperature of 15 °C.

Using the DSR, the magnitude of the $|G^*|$ and $\delta$ were measured at 60 °C at low strain, unaged, and after RTFO and subsequent PAV aging. The aging index serves as a quantitative indicator, characterizing the extent of the changes in the rheological parameters and, thereby, describing the aging degree of the asphalt binder. This index is derived from the ratio of both values, with a lower index number representing less susceptibility to aging. These indexes are calculated using Equations (1) and (2).

$$\text{AgingIndex}(|G^*|) = \frac{|G^*_{\text{aged}}|}{|G^*_{\text{unaged}|}} \tag{1}$$

$$\text{AgingIndex}(\delta) = \frac{\delta_{\text{unaged}}}{\delta_{\text{aged}}} \tag{2}$$

The Glover-Rowe (G-R) parameter is a valuable tool for assessing the cracking resistance of asphalt binders based on their complex modulus and phase angle at low temperatures [49,50]. It is computed using Equation 3 and can also evaluate the effects of aging on asphalt binders, as aging tends to increase the parameter and reduce the cracking resistance of the binder. The G-R parameter's application extends to optimizing asphalt mix designs and the development of cost-effective, high-performance asphalt materials [49]. In this work, the G-R was determined for three different aging conditions of each sample, RTFO-aged, RTFO + 20 h PAV-aged, and RTFO + 40 h PAV-aged, to analyze the progression of aging.

$$G - R = \frac{G^*(\cos\delta)^2}{\sin\delta} \tag{3}$$

The asphalt oxidation reaction has, as a byproduct, the production of functional groups, including carboxylic acids and sulfoxides [34–36]. The formation of carbonyl (C=O) and sulfoxide (S=O) groups during the aging process affects asphalt hardening, making them reliable indicators of aging [51,52]. These chemical bonds can be identified by analyzing the absorbance intensities of infrared using Fourier-transform infrared spectroscopy (FTIR) [53–55]. Asphalt FTIR spectra, tested using Thermo Scientific Nicolet iS5 FTIR, provide valuable information about the functional groups and molecular vibrations present in the material. These spectra can reveal details about the asphalt's chemical structure, essential for understanding its properties and behavior. In the spectra, the stretching vibration of C=O is around 1700 cm$^{-1}$, and around 1000 cm$^{-1}$ for S=O. The aging process focused on wavenumbers between 600 and 2000 cm$^{-1}$, a region chosen to be the base for

the calculation of the carbonyl index (Ic=o) and sulfoxide index (Is=o). Those indexes were calculated according to Equations 4 and 5, respectively.

$$I_{C=O} = \frac{\text{Area of spectral bands between 1650 and 1747 cm}^{-1}}{\text{Area of spectral bands between 600 and 2000 cm}^{-1}} \tag{4}$$

$$I_{S=O} = \frac{\text{Area of spectral bands between 993 and 1046 cm}^{-1}}{\text{Area of spectral bands between 600 and 2000 cm}^{-1}} \tag{5}$$

## 3. Results

As shown in Table 1, modifying the neat binder with polymer improved the high-temperature PG and the elasticity of the binder, improving recovery and decreasing Jnr, a parameter that is related to rutting susceptibility. Furthermore, the addition of sulfur, up to 0.3% by the weight of asphalt, amplified these enhancements, elevating the PMA from being suitable for very heavy traffic (>10 million ESAL's) to effectively supporting extremely heavy traffic (>30 million ESAL's) loads.

**Table 1.** Binder PG and MSCR results.

| Sample | PG | MSCR @ 64 °C | | |
| --- | --- | --- | --- | --- |
| | | Jnr [1/kPa] | %Recovery | Grade |
| C | 64–22 | 1.99 | 0.88 | S |
| C + SB | 70–22 | 0.68 | 22.05 | V |
| C + 0.075 S | 64–22 | 1.57 | 2.010 | H |
| C + SB + 0.03 S | 76–22 | 0.39 | 45.62 | E |
| C + SB + 0.075 S | 76–22 | 0.49 | 37.59 | E |
| C + SB + 0.1 S | 76–22 | 0.48 | 37.69 | E |
| C + SB + 0.3 S | 76–22 | 0.21 | 65.63 | E |
| C + SB + 0.5 S | 76–18 | 0.67 | 21.11 | Fail |

Figure 2 visually illustrates the outcomes of the polymer modification and introduction of sulfur into PMA. The control sample, as expected, has a low %recovery and does not meet the minimum value for its Jnr when tested at 3.2 kPa shear stress, according to the AASHTO TP 70 [46] criteria, indicating that the binder is not polymer-modified. Only the samples of PMA with added sulfur (0.03, 0.075, 0.1, and 0.3%) have a significant elastic response, which suggests that the presence of sulfur helps the polymer modification efficiency in terms of elastic recovery.

Based on the Christensen–Anderson model and Williams–Landel–Ferry (WLF) time–temperature superposition principle, the master curves for complex modulus (G*) and phase angle (δ) were constructed using 15 °C as a reference temperature and are shown in Figure 3, in which they represent the average of three replicates. As expected, G* and δ exhibit frequency dependence; with increasing frequency, G* rises while δ decreases. In the low-frequency region of the master curves, introducing polymer into the asphalt binder led to an increase in G* and a decrease in δ, ideal for preventing permanent deformation. However, when sulfur was subsequently added, the binder experienced a slight reduction in stiffness, although it remained stiffer than the unmodified binder, and there was no observable change in its elasticity. In the high-frequency range, the incorporation of polymer reduced the stiffness and phase angle of the binder, making it less susceptible to cracking. The subsequent addition of sulfur led to a slight increase in G* and made the binder marginally more viscous than the PMA. Nevertheless, it is essential to highlight that despite these alterations, the PMA with sulfur still outperformed the unmodified base binder.

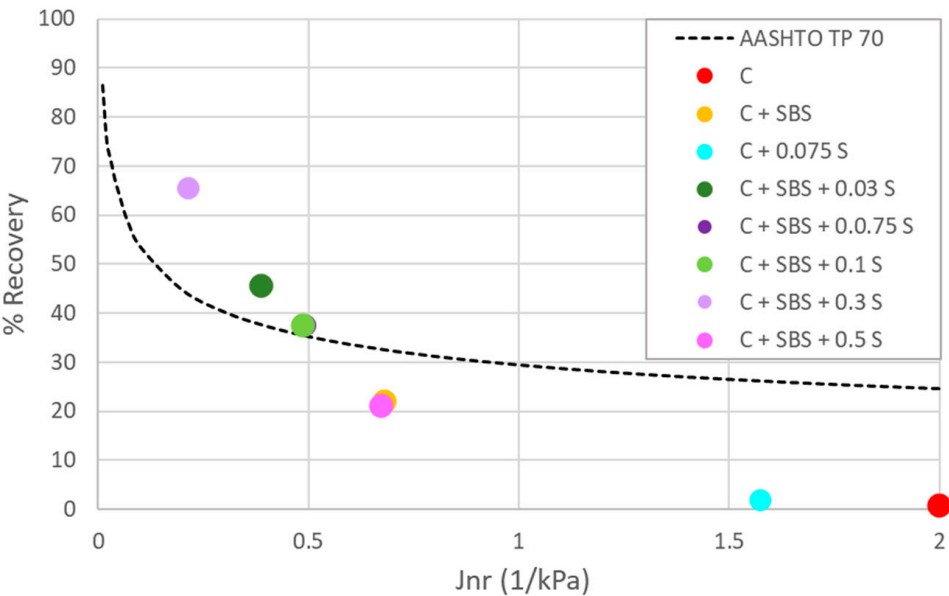

**Figure 2.** Elastomeric behavior of PMA.

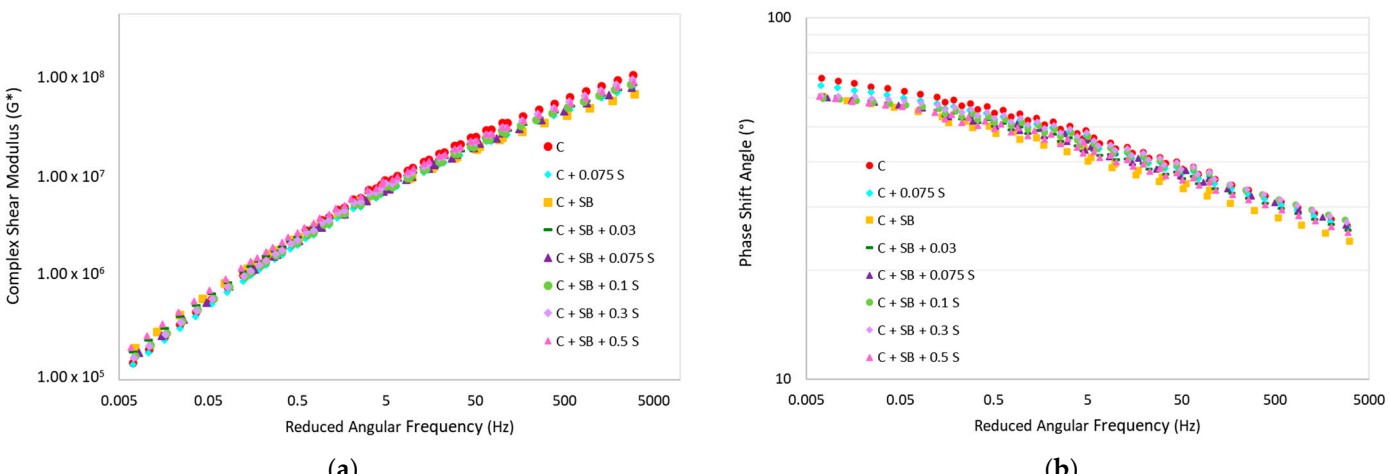

(**a**)　　　　　　　　　　　　　　　　　　　　(**b**)

**Figure 3.** Master curves' (**a**) complex shear modulus; (**b**) phase shift angle.

The impact of sulfur as a delaying agent for PMA aging is observable through an analysis involving the Glover-Rowe parameter and aging indexes. As the sample undergoes aging, it becomes increasingly brittle and susceptible to cracking, a phenomenon reflected in the rising values of the G-R parameter, as shown in Figure 4. Short-term aging impacts the G-R parameter, but it is the extended service life, simulated through one or two 20 h PAV-aging cycles, that substantially increases the likelihood of cracking occurrence. The initial modification of the binder with polymer increased its susceptibility to cracking after short- and long-term aging, and adding sulfur up to 0.3% could reduce the effects of aging on PMA. The parameter exhibited its highest value at 0.5% wt. of sulfur in PMA, indicating that beyond a certain threshold, an excess of sulfur can actually detrimentally impact aging susceptibility.

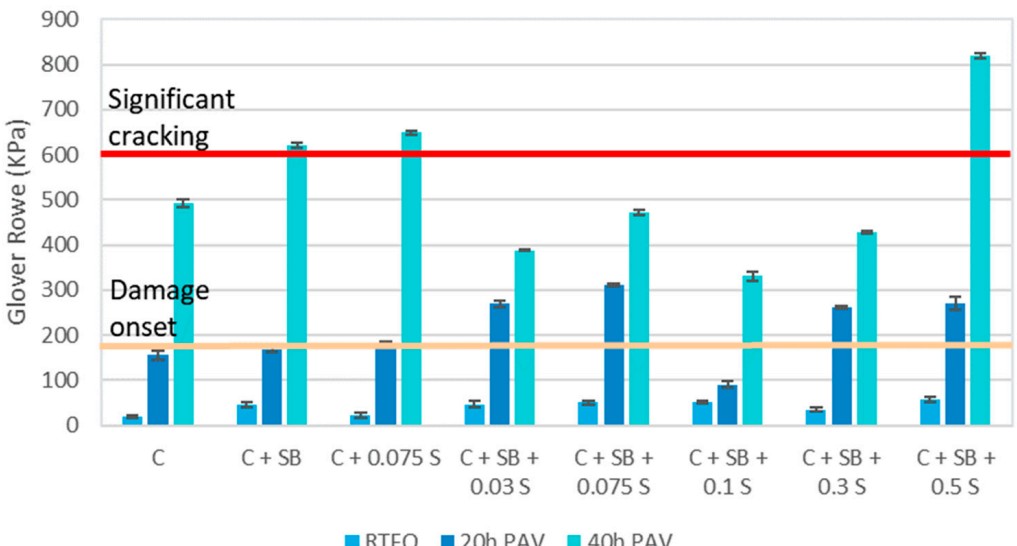

**Figure 4.** Glover-Rowe parameter and cracking susceptibility.

Figure 5 illustrates the impacts of aging on the binder's complex modulus (G*) and phase angle as an aging index. This index represents the ratio between the rheological parameters before and after the aging process. A lower value indicates less susceptibility to aging. It is therefore inferred that adding sulfur to the asphalt–polymer blend, up to 0.3% wt., effectively mitigates the aging susceptibility of PMA, as is evident from the notable reduction in the aging indexes. Furthermore, it is crucial to underscore the critical role that the quantity of sulfur assumes in directing these alterations. The most optimal formulation emerges at a sulfur content of 0.1%, in agreement with the Glover-Rowe parameter analysis.

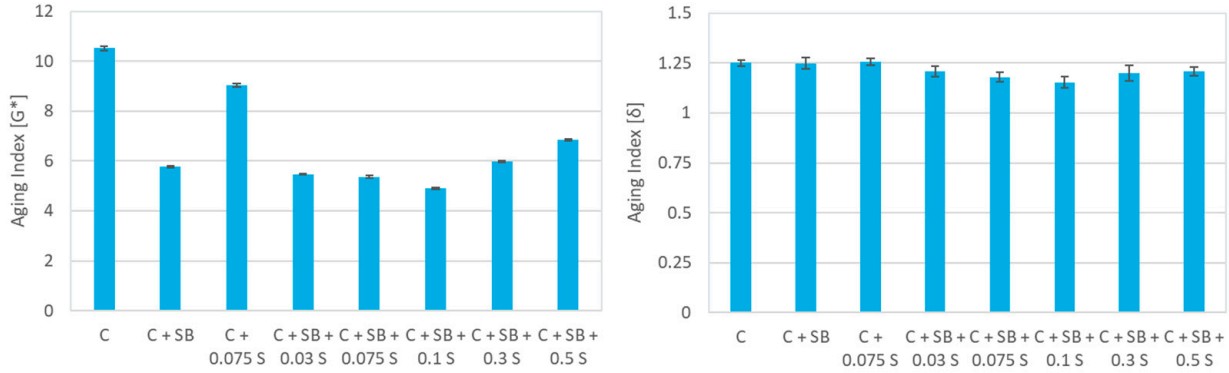

**Figure 5.** Aging indexes (G* and δ).

The FTIR spectra of the PMA samples with the addition of sulfur are shown in Figure 6. The trends for all the samples were similar and typical of an asphalt binder, indicating that no new functional group was generated with the different formulations.

Figure 7 presents the calculated values of Ic=o and Is=o for the PAV-aged samples, as determined by Equations (4) and (5). However, a consistent trend is observed across all samples, indicating that no new functional groups were generated as a result of the different formulations, and no significant change was observed. Thus, there is no reason to conclude that sulfur accelerates or mitigates the formation of carbonyls and sulfoxides.

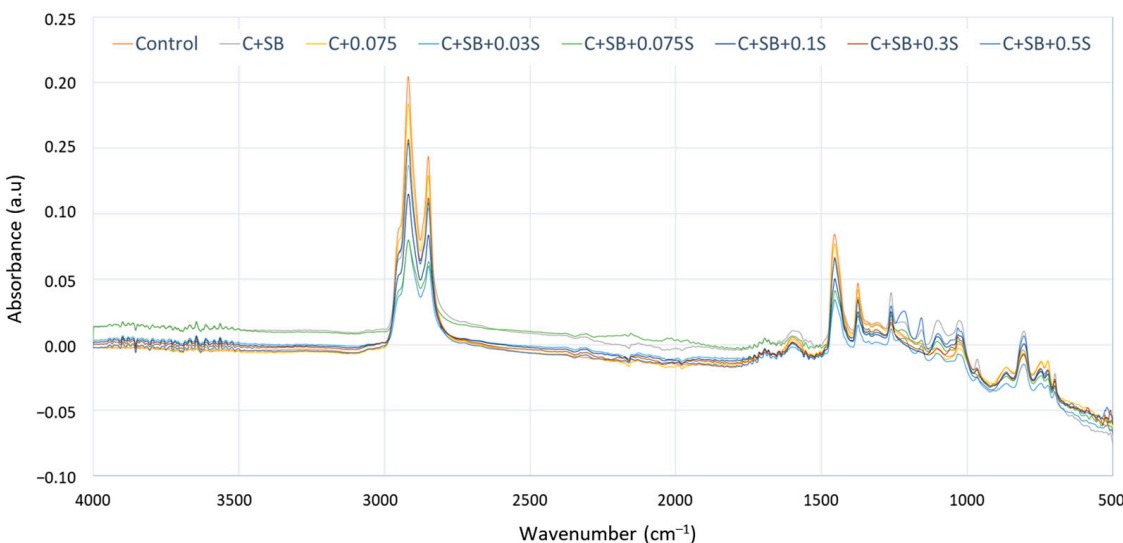

**Figure 6.** FTIR spectra of asphalt binder with different sulfur contents.

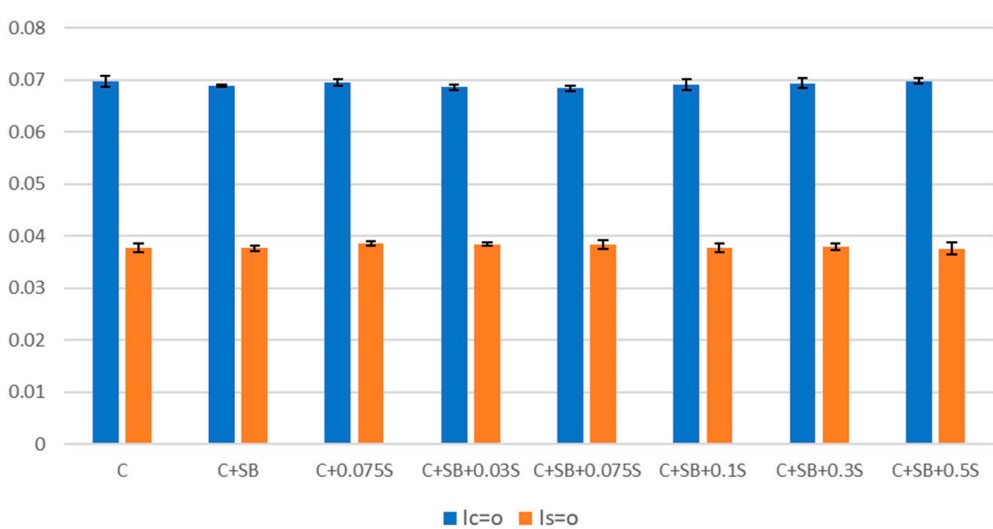

**Figure 7.** Ic=o and Is=o values of asphalt samples.

## 4. Conclusions

In conclusion, this research has provided valuable insights into the impact of adding sulfur on the rheological properties and aging mechanisms of PMA. Through a comprehensive analysis involving PG, elastomeric behavior, complex shear modulus and phase angle master curves, the Glover-Rowe parameter, aging indexes, and carbonyl and sulfoxide indexes, we have observed a significant improvement in the rheological properties and anti-aging potential of sulfur in PMA.

Specifically, it has been found that sulfur content has a substantial and positive effect, particularly when added at up to 0.3% of the total weight of the binder. This addition enhances the performance grade, elevating it by one grade of the high-temperature PG classification, and improves the elastomeric behavior, enabling the PMA to meet the recovery criteria and support heavier traffic, all without compromising its resistance to low-temperature cracking. The inclusion of sulfur has not had a notable effect on the carbonyl and sulfoxide indexes. Furthermore, a reduction in aging susceptibility indexes and the Glover-Rowe parameter suggests that sulfur effectively retards the aging process of SBS polymer-modified asphalt, reducing the aging potential by approximately 15% when considering these parameters.

It is crucial to continue investigating the practical implementation and broader implications of PMA with the addition of sulfur. Future research should focus on long-term field performance evaluations to validate laboratory findings, economic, environmental, and health impact assessments, and optimizing the sulfur content for various base asphalt binders.

**Author Contributions:** Conceptualization, A.L.R. and R.C.W.; methodology, A.L.R., C.F. and R.C.W.; validation, A.L.R., C.F. and R.C.W., formal analysis, A.L.R.; investigation, A.L.R. and C.F.; resources, A.L.R.; data curation, A.L.R.; writing—original draft preparation, A.L.R.; writing—review and editing, A.L.R., C.F. and R.C.W.; visualization, A.L.R.; supervision, R.C.W.; project administration, A.L.R. All authors have read and agreed to the published version of the manuscript.

**Funding:** This research received no external funding.

**Institutional Review Board Statement:** Not applicable.

**Informed Consent Statement:** Not applicable.

**Data Availability Statement:** No new data were created or analyzed in this study. Data sharing is not applicable to this article.

**Conflicts of Interest:** The authors declare no conflict of interest.

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
