# Peer review of "Rheological and Aging Characteristics of Polymer-Modified Asphalt with the Addition of Sulfurâ€"

_infrastructures, doi:10.3390/infrastructures8110160_

Round 1
Reviewer 1 Report
Comments and Suggestions for Authors
Rheological and aging characteristics of polymer-modified asphalt with addition of sulfur
Comments:
This work studied the influence of SBS polymer and sulfur on the rheological and aging properties of asphalt. This topic is not new, but it would be beneficial to industrial engineering. Here are some comments provided for further improving this manuscript.
(1) The research gap and objectives have to be clarified in abstract. Why it is meaningful or interesting to study the sulfur influence on SBS modified bitumen?
(2) English editing has to be checked carefully, especially for the tense issue. (For example, in abstract, the results indicated that…)
(3) Introduction: more state-of-the-art references can be considered. Toward the long-term aging influence and novel reaction kinetics models of bitumen. International Journal of Pavement Engineering. Storage stability and its relationship with microstructure of SBS modified de-oiled asphalt. The continuous swelling-degradation behaviors and chemo-rheological properties of waste crumb rubber modified bitumen considering the effect of rubber size. Construction and Building Materials.
(4) More basic properties of bitumen and polymer are expected in section 2.1.
(5) It was mentioned in Figure 1 that the polymer modified asphalt will be prepared for 5 hours totally. Why select the 4 hours in the second step. As the review know, the long preparation time would lead to the aging of polymer modified asphalt.
(6) The unit of Jnr in Table 1 should be added. Check the same issue in all tables.
(7) The error bar can be added in Figures 4, 5, and 7.
(8) The equations for the aging index calculations should be mentioned.
(9) Some recommendations for future work can be supplemented.
Comments on the Quality of English LanguageEnglish editing has to be checked carefully, especially for the tense issue. (For example, in abstract, the results indicated that…)
Author Response
Hi,
Thank you so much for taking the time to review this paper and provide insightful comments. I'm sending the attached paper with the corrections. I have marked with comments the changes made to address your questions and suggestions. I truly believe they helped to improve my paper.

Reviewer 2 Report
Comments and Suggestions for Authors
This paper studied the effect of sulfur addition on the rheological properties and aging mechanism of polymer-modified asphalt. The topic of the article is relevant at this time. The article is easy to read, the graphs and results are clearly described. There are the following comments and questions about the work:
Why did you choose 3% SB, and not more, for example, 7%?
The numbering under the pictures and in the text is mixed up.
Line 7-10 “methods”: Were the samples aged first in RTFO, then in PAV? Or were the samples tested after each oven/vessel exposure? Then why are the results of aging studies only for one aging method?
Author Response
Dear reviewer,
We would like to express our sincere gratitude for taking the time to review our research paper and for providing valuable comments and feedback to enhance the quality of our work.
(1) Why did you choose 3% SB, and not more, for example, 7%?
We chose to work with 3% SBS polymer in our study to align with the typical industry standards and practices, and the literature. Additionally, 7% is generally reserved for highly modified asphalt, and our research aimed to investigate the effects of sulfur addition on a more common and practical polymer content.
(2) The numbering under the pictures and in the text is mixed up.
We appreciate your keen observation regarding the numbering under the pictures and its alignment with the text. We reviewed and corrected those issues to ensure consistency and clarity in our paper.
(3) Line 7-10 “methods”: Were the samples aged first in RTFO, then in PAV? Or were the samples tested after each oven/vessel exposure? Then why are the results of aging studies only for one aging method?
Thank you for your comment and for pointing out the need for clarification regarding the aging methods. To clarify, the samples were indeed aged sequentially in both the Rolling Thin Film Oven (RTFO) and the Pressure Aging Vessel (PAV). The results are for the RTFO + PAV -aged samples. We made changes in the Methods section, to make this clear to the reader.
I have attached the revised paper, which includes the necessary revisions made in response to your comments and those of the other reviewers. Thank you once again for your feedback, it has been invaluable in improving the quality of the paper.
Sincerely,
Ana Rodrigues
Reviewer 3 Report
Comments and Suggestions for Authors
This study aims to investigate the influence of chemical cross-linkers such as sulfur on the rheological, and aging characteristics of polymer-modified asphalt (PMA) binder. Some conclusions were obtained. The following comments are made:
1. In the section of the Introduction, please point out the gaps between the previous research and the study in this paper.
2. In 2.1 materials, please show the reasons for the selection of the content, such as (0.075, 0.03, 0.1, 0.3, 0.5 wt.%) of sulfur, 160°C and 800 rpm for 4 hours, etc.
3. “The FTIR spectra of the PMA samples with the addition of sulfur are shown in Figure 5”, please check the number of the Figure. It is recommended to give more explanations on the results of FTIR.
Comments on the Quality of English LanguageMinor editing of English language required.
Author Response
Dear reviewer,
We wish to extend our heartfelt appreciation to the esteemed reviewer for generously dedicating their valuable time to evaluate our paper and offering us invaluable feedback. Your profound insights have played a pivotal role in enhancing the quality of our research, and we genuinely value your diligent efforts and expertise in critiquing our work.
(1) In the section of the Introduction, please point out the gaps between the previous research and the study in this paper.
We extend our sincere appreciation to the reviewer for their insightful suggestion. In response to the request to highlight the gaps between prior research and the current study in the Introduction, we have incorporated the following passage to address this:
"While some valuable research has addressed aspects of PMA and sulfur vulcanization to enhance its storage stability, there remain notable gaps in the comprehensive characterization of improvements in rheological properties and anti-aging potential when sulfur is incorporated into asphalt-polymer blends. Despite individual studies, such as Cuciniello [39] and Tang [10], that have indicated that crosslinked polymer-modified binders exhibited a lower oxidative susceptibility than non-cross-linked binders, a holistic understanding of these critical aspects is currently lacking."
This addition helps to emphasize the distinctive contribution of our research and highlights the unexplored dimensions in the field. We are grateful for your guidance in strengthening our paper.
(2) In 2.1 materials, please show the reasons for the selection of the content, such as (0.075, 0.03, 0.1, 0.3, 0.5 wt.%) of sulfur, 160°C and 800 rpm for 4 hours, etc.
We greatly appreciate the reviewer's comment and their attention to the details of our study. The selection of the specific ratios for sulfur content (0.075, 0.03, 0.1, 0.3, 0.5 wt.%) was made after careful consideration and pre-screening tests to ensure their relevance to our research. The ratios were also consistent with findings from relevant literature that studied sulfur as a crosslinker in asphalt. The temperature of 160°C was adopted to simulate storage tank conditions and facilitate effective curing. The study "Investigation on Preparation Method of SBS-Modified Asphalt Based on MSCR, LAS, and Fluorescence Microscopy." recommends 6 hours of mixing duration to certify the sulfur SB-vulcanization. A mixing duration of 4 hours was employed in accordance with FHWA recommendations, and we ensured not to exceed this duration.
(3) “The FTIR spectra of the PMA samples with the addition of sulfur are shown in Figure 5”, please check the number of the Figure. It is recommended to give more explanations on the results of FTIR.
We have corrected the numbering for Figure 5 as per their suggestion. Additionally, we have made enhancements to provide more detailed explanations regarding the FTIR results in the revised manuscript.
We would like to reiterate our gratitude for your valuable input. I have attached the revised paper, which includes the necessary revisions made in response to your comments and those of the other reviewers, and with comments highlighting the changes I've made addressing your points. Your comments have significantly enhanced the clarity and comprehensiveness of our paper, ensuring that it meets the high standards of academic excellence.
Sincerely,
Ana Rodrigues
Reviewer 4 Report
Comments and Suggestions for Authors
Comments to authors
This study explored the possibilities of incorporating sulfur in order to assess the changes in the aging and rheological performance of polymer-modified bitumen. It is a well-written paper, however, the results and discussion must be improved and a few other remarks as detailed comments are provided below should be addressed.
· Abstract
o The abstract is the most read part of a paper, Hence, it can be improved by providing some major findings in percentages instead of mentioning the PMA performance can be improved using sulfur.
· Introduction
o This section can be improved by providing more info regarding sulfur addition to the bitumen. Besides, the first sentence should be checked and the second paragraph should not have an extra dot.
· Materials and methods
o Why the amount of 3% SB was used? Please justify
o Can the author explain why the mixing shear rate (4000 RPM) in the text is different compared to the value provided in Figure 1 (3000 RPM)? A more detailed explanation of the mixing parameters can be an added value to the paper.
o It is recommended that the authors explain how the master curves were plotted.
o The number of test replicates is missing
· Results and discussion
o It is stated that only the PMA with the addition of sulfur achieves the minimum % Recovery. Does it mean that other samples even pure and polymer-modified ones cannot meet the requirements? More explanation would be an added value.
o The number of Figures is not correct and should be fixed both in the text and captions.
o More explanation should be provided for the master curves’ Figure.
o No explanation for the changes in the trend of short term aging can be found. In addition, no discussion or comparison between changes in the trend of Glover-Rowe subjected to different aging can be found in the text.
o What aging condition (RTFO or PAV) and G* testing parameters were used to define the aging index?
· Conclusions
o There is a contradiction between the conclusion and the previous section. The authors mentioned no obvious change can be found from FTIR while in the conclusion they explained it differently.
o How did the authors know that the impact of sulfur is significant without any statistical analysis?
o The conclusion is very general and should be improved
Comments on the Quality of English LanguageSatisfactory
Author Response
Dear Reviewer,
We want to extend our gratitude for your thorough evaluation of our paper and for providing constructive feedback. Your insights have been invaluable in improving the quality and clarity of our research.
We have taken your comments into careful consideration and made the necessary revisions to address the issues you raised. The changes have been marked in the attached document. Below is a summary of the modifications made in response to your comments:
-
Abstract:
- We have enhanced the abstract by incorporating specific figures-based findings to better convey the outcomes of our study, e.g. one bump in the PG grade, increased traffic level, 15% reduce. -
Introduction:
- The introduction has been revised to provide more detailed information regarding sulfur, the addition of sulfur to bitumen, aging mechanisms of asphalt binder, and sulfur as a crosslinker. -
Materials and Methods:
- We chose to work with 3% SB polymer in our study to align with the typical industry standards and practices, and to align with findings from relevant literature about sulfur as a asphalt crosslinker.
- The discrepancy in mixing shear rates has been rectified, and we have added a more comprehensive explanation of the mixing parameters.
- We've included an explanation of how the master curves were plotted using the Christensen-Anderson model, and the shift factor was calculated based on the Williams-Landel-Ferry (WLF) time-temperature superposition principle for a reference temperature of 15 â—¦C. -
Results and Discussion:
-We have provided a more detailed explanation regarding the % Recovery and its implications.
- The issue with the number of figures has been corrected
- We have expanded on the master curves' interpretation and results.
- We haven't analyzed short-term aging. We clarified in the paper that we analyzed samples that were aged in RTFO+PAV.
- The aging conditions and G* testing parameters used to define the aging index have been clarified. -
Conclusions:
- We've resolved the contradiction between the conclusion and the previous section by aligning the statements regarding FTIR results.
- We added statistical analysis, as error bars, in Figures 4, 5, and 7, to support the significance of sulfur's impact
- The conclusion has been revised to provide a more specific summary.
We sincerely appreciate your feedback and the time you dedicated to reviewing our paper. Your input has undoubtedly strengthened the quality of our research, and we hope the revisions address your concerns adequately.
Kindly find the marked changes with comments in the attached document. Your continued support is greatly valued.
Best regards,
Ana Rodrigues
Round 2
Reviewer 1 Report
Comments and Suggestions for Authors
Most my comments have been considered well in revised manuscript. Thanks!